# Target Localization Based on High Resolution Mode of MIMO Radar with Widely Separated Antennas

**Jiaxin Lu** [1,2]![ID], **Feifeng Liu** [1,2,*], **Hongjie Liu** [1,2] **and Quanhua Liu** [1,2]

1   School of Information and Electronics, Beijing Institute of Technology, Beijing 100081, China;
    3120170413@bit.edu.cn (J.L.); 3120200792@bit.edu.cn (H.L.); liuquanhua@bit.edu.cn (Q.L.)
2   Key Laboratory of Electronic and Information Technology in Satellite Navigation (Beijing Institute of
    Technology), Ministry of Education, Beijing 100081, China
*   Correspondence: feifengliu_bit@bit.edu.cn

**Abstract:** Coherent processing of multiple-input multiple-output (MIMO) radar with widely separated antennas has high resolution capability, but it also brings ambiguity in target localization. In view of the ambiguity problem, different from other signal processing sub-directions such as array configuration optimization or continuity of phase in space/time, this paper analyzes it from the information level, that is, the tracking method is adopted. First, by using the state equation and measurement equation, the echo data of multiple coherent processing intervals (CPI) are collected to improve the target localization accuracy as much as possible. Second, the non-coherent joint probability data association filter (JPDAF) is used to achieve stable tracking of spatial cross targets without ambiguity measurements. Third, based on the tracking results of the non-coherent JPDAF, the ambiguity of coherent measurement is resolved, that is, the coherent JPDAF is realized. By means of non-coherent and coherent alternating JPDAF (NCCAF) algorithms, high accuracy localization of multiple targets is achieved. Finally, numerical simulations are carried out to validate the effectiveness of the proposed NCCAF algorithm.

**Keywords:** MIMO radar; high resolution mode; JPDAF; parameter estimation; multi-target

## 1. Introduction

According to [1,2], the coherent processing of multiple-input multiple-output (MIMO) radar with widely separated antennas can locate targets with high resolution. However, the coherent processing of widely distributed antennas has grating lobes. How to realize phase unwrapping is key to achieving high resolution target location in MIMO radar with widely separated antennas. Due to the remarkably high resolution of coherent processing, phase unwrapping exists in all sub-directions of the signal processing field.

In array signal processing, array position arrangement can be understood as spatial sampling. For linear array, there will be no grating lobes as long as the antenna interval is half of the wavelength. Similar to the redundancy of equally spaced sampling in time domain [3], minimum-redundancy arrays [4–6], nested arrays [7], and co-prime arrays [8–10] have been proposed successively. However, the above arrays do not consider the prior information such as non-coherent processing, and only realize the array configuration optimization from the perspective of spatial snapshot. Therefore, the optimized array configuration still has half-wavelength array spacing. In other words, it is not suitable for the phase unwrapping of widely distributed antennas. Ref. [11] assumes that the antenna beam of radar nodes is narrow, so as to ensure that the antenna interval is much larger than the wavelength. However, this algorithm also brings the beam synchronization problem of radar nodes.

In SAR imaging or image field, each pixel is surrounded by other pixels, that is, the phase is continuous in space. Therefore, methods such as parameter estimation [12,13], clustering [14], extended minimum cost flow [15], and neural network [16–18] can be used

for phase unwrapping. In the field of acoustic signal processing, because the signal carrier frequency is low and the relative speed of target and sensor is small, phase unwrapping can be realized through the continuity of phase in time [19,20]. However, MIMO radar with widely separated antennas do not have high precision imaging in the process of target location. In addition, MIMO radar has a high carrier frequency relative to the acoustic signal, and the relative velocity of target and radar is high. Hence, the phase unwrapping cannot be realized through the phase space-time continuity mentioned above.

In the field of one-dimensional high resolution range profile (HRRP), the phase-derived range (PDR) is developed by taking the envelope ranging as a priori information and then realizing the phase unwrapping [21,22]. The success of PDR is based on the root mean square error (RMSE) of envelope ranging multiplied by a carrier frequency less than one. According to Cramér-Rao bound (CRB) of delay estimation, RMSE of envelope ranging is determined by signal bandwidth and signal-to-noise ratio (SNR). Therefore, the design of large bandwidth signal and long-term coherent integration is the key of PDR [23]. Refs. [24,25] realize PDR by stepped-frequency chirp signal. Ref. [26] carries out the phase-derived velocity (PDV) of a high speed target. However, the above work does not take advantage of widely distributed antennas, that is, the coherent processing is not carried out based on the non-coherent target location results.

Based on a tracking method [27–29] and the motion characteristics of the target, this paper realizes phase unwrapping from the information level. Firstly, based on the non-coherent target location results, joint probability data association filter (JPDAF) is used to ensure the correct correlation of multiple targets in spatial crossing. Non-coherent target location results and coherent target location results are obtained at the same time, so the effective use and combination of the two measurements is important. Therefore, based on the association filtering results of non-coherent JPDAF, JPDAF is used again to select the correct measurement from the coherent ambiguous measurements for tracking. By using non-coherent and coherent alternating JPDAF (NCCAF) algorithm, high resolution target location in multi-target scenarios is achieved.

The rest of the paper is organised as follows: Section 2 first establishes the signal model from information level, and then NCCAF algorithm is derived. Section 3 verifies the feasibility of the algorithm through numerical simulations. Section 4 draws the conclusion.

## 2. Materials and Methods

### 2.1. Signal Model

Let the state equation of target motion be:

$$\boldsymbol{x}^q(k) = \mathbf{F}(k-1)\boldsymbol{x}^q(k-1) + \boldsymbol{\Gamma}(k-1)\boldsymbol{v}^q(k-1), \tag{1}$$

where $k$ represents the sampling time. $\boldsymbol{x}^q(k)$ represents the state vector of the $q$th target at time $k$, which can be written as $\boldsymbol{x}^q(k) = [x^q(k), \dot{x}^q(k), \ddot{x}^q(k), y^q(k), \dot{y}^q(k), \ddot{y}^q(k)]^T$. $(x^q(k), y^q(k))$ is the target position. $(\dot{x}^q(k), \dot{y}^q(k))$ is the target velocity. $(\ddot{x}^q(k), \ddot{y}^q(k))$ is the target acceleration. $\boldsymbol{v}^q(k-1)$ is the white Gaussian noise vector with the mean of $\mathbf{0}$, and its covariance matrix is $\mathbf{Q}(k-1)$. $\mathbf{F}(k-1)$ and $\boldsymbol{\Gamma}(k-1)$ are state transition matrix and process noise distribution matrix, respectively. They can be written as:

$$\mathbf{F}(k-1) = \begin{bmatrix} 1 & T & T^2/2 & 0 & 0 & 0 \\ 0 & 1 & T & 0 & 0 & 0 \\ 0 & 0 & 1 & 0 & 0 & 0 \\ 0 & 0 & 0 & 1 & T & T^2/2 \\ 0 & 0 & 0 & 0 & 1 & T \\ 0 & 0 & 0 & 0 & 0 & 1 \end{bmatrix}, \tag{2}$$

$$\mathbf{\Gamma}(k-1) = \begin{bmatrix} T^2/2 & 0 \\ T & 0 \\ 1 & 0 \\ 0 & T^2/2 \\ 0 & T \\ 0 & 1 \end{bmatrix}, \tag{3}$$

where $T$ is the interval of sampling time.

The measurement equation of the target in non-coherent scenario is:

$$z_s^q(k) = \mathbf{H}(k)x^q(k) + w_s^q(k), \tag{4}$$

where $z_s^q(k)$ is the measurement in non-coherent scenario; $w_s^q(k)$ is assumed to be white Gaussian noise with a mean of $\mathbf{0}$, and its covariance matrix is assumed to be $\mathbf{R}_s^q(k)$; $\mathbf{H}(k)$ is the measurement matrix, which can be written as:

$$\mathbf{H}(k) = \begin{bmatrix} 1 & 0 & 0 & 0 & 0 & 0 \\ 0 & 0 & 0 & 1 & 0 & 0 \end{bmatrix}. \tag{5}$$

The measurement equation of the target in coherent scenario is:

$$z_u^q(k) = \mathbf{H}(k)x^q(k) + w_u^q(k), \tag{6}$$

where $z_u^q(k)$ is the measurement in coherent scenario; $w_u^q(k)$ is white Gaussian noise with a mean of $\mathbf{0}$, and its covariance matrix is $\mathbf{R}_u^q(k)$. The first difference between non-coherent measurement $z_s^q(k)$ and coherent measurement $z_u^q(k)$ is noise variance. According to CRB [2], the RMSE of time delay estimation in coherent scenario is:

$$\text{RMSE}_u(\tau^q) = \frac{1}{8\pi^2 SNR(\beta_k^2 + f_c^2)}, \tag{7}$$

where $\beta_k$ is the effective bandwidth; $f_c$ is the carrier frequency; $\tau^q$ is the time delay of the $q$th target. However, in the non-coherent scenario, the delay information at carrier frequency is ignored. Hence, the RMSE of time delay estimation in non-coherent scenario can be written as:

$$\text{RMSE}_s(\tau^q) = \frac{1}{8\pi^2 SNR\beta_k^2}. \tag{8}$$

By comparing the above two equations, it can be seen that $\mathbf{R}_u^q(k)$ is smaller than $\mathbf{R}_s^q(k)$, thus ensuring the high accuracy of coherent measurements. At the same time, because the echo delay is estimated from $e^{(-j2\pi f_c \tau^q)}$ in the coherent scenario, there are many ambiguous measurements. Therefore, the complete coherent measurements can be written as $z_{uj}(k)$, where $j = 1, \ldots, m_{uk}$. $m_{uk}$ represents the number of coherent measurements. Let the number of targets be $Q$. Because of the existence of ambiguous measurements, it is clear that $m_{uk} > Q$. For MIMO radar with widely separated antennas, the coherent measurements and non-coherent measurements of the targets are obtained simultaneously. Then, at time $k$, the total measurements of radar can be set as $\mathbf{Z}(k) = \{\mathbf{Z}_u(k), \mathbf{Z}_s(k)\}$, where $\mathbf{Z}_u(k)$ is the set of coherent measurements, and $\mathbf{Z}_s(k)$ is the set of non-coherent measurements.

### 2.2. NCCAF Algorithm

As can be seen in Section 2.1, the number of coherent measurements is $m_{uk}$. In the multi-target tracking scenario, how to select the correct measurement from the ambiguous measurements to associate with the target is the problem to be solved in the subsection. The problem is illustrated in Figure 1.

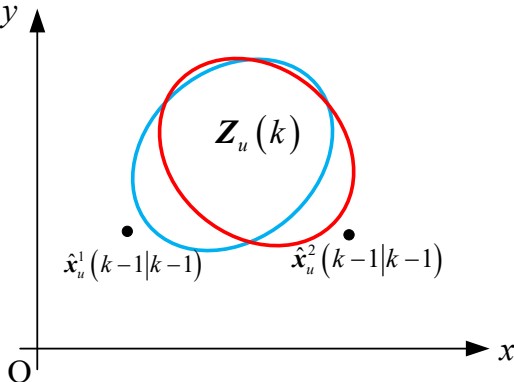

**Figure 1.** The scenario of multiple targets in spatial crossing and ambiguous measurements.

In Figure 1, the blue curve represents the distribution range of the coherent ambiguous measurements of the first target; the red curve represents the distribution range of the coherent ambiguous measurements of the second target. For the radar system, all measurements will be collected without distinguishing which target they belong to, that is, $Z_u(k)$.

### 2.2.1. Non-Coherent JPDAF

Since there is no ambiguity in the non-coherent measurements, this paper first uses non-coherent measurements to assist the coherent measurements. Because the non-coherent and coherent alternating algorithm is adopted in this paper, at time $k-1$, the estimation of the $q$th target state is known as $\hat{x}_u^q(k-1|k-1)$ and the corresponding covariance matrix is $P_u^q(k-1|k-1)$. The further prediction of state vector and the corresponding covariance matrix are:

$$
\begin{aligned}
\hat{x}_u^q(k|k-1) &= \mathbf{F}(k-1)\hat{x}_u^q(k-1|k-1), \\
\mathbf{P}_u^q(k|k-1) &= \mathbf{F}(k-1)\mathbf{P}_u^q(k-1|k-1)\mathbf{F}^T(k-1) + \mathbf{\Gamma}(k-1)\mathbf{Q}(k-1)\mathbf{\Gamma}^T(k-1),
\end{aligned}
\tag{9}
$$

respectively. Then, the further prediction of measurement vector and the corresponding covariance matrix are:

$$
\begin{aligned}
\hat{z}_u^q(k|k-1) &= \mathbf{H}(k)\hat{x}_u^q(k|k-1), \\
\mathbf{S}_s^q(k) &= \mathbf{H}(k)\mathbf{P}_u^q(k|k-1)\mathbf{H}^T(k) + \mathbf{R}_s^q(k),
\end{aligned}
\tag{10}
$$

respectively. The corresponding gain matrix is:

$$
\mathbf{K}_s^q(k) = \mathbf{P}_u^q(k|k-1)\mathbf{H}^T(k)\left(\mathbf{S}_s^q(k)\right)^{-1}.
\tag{11}
$$

Let the number of non-coherent measurements be $m_{sk}$. In order to realize the association between non-coherent measurements and targets, Bar–Shalom proposed the validation matrix [30]. The validation matrix can be written as:

$$
\mathbf{\Omega}_s = \begin{bmatrix} \omega_1^1 & \cdots & \omega_1^Q \\ \vdots & \cdots & \vdots \\ \omega_{m_{sk}}^1 & \cdots & \omega_{m_{sk}}^Q \end{bmatrix},
\tag{12}
$$

where $\omega_j^q$ indicate whether the $j$th measurement falls into the validation gate of the $q$th target. Validation gate refers to the area where the correct measurement of the target is likely to occur, and this area center is the prediction of the measurement, that is, $\hat{z}_u^q(k|k-1)$. When $\omega_j^q = 1$, it means $z_{sj}(k)$ can associate with the $q$th target; when $\omega_j^q = 0$, it means $z_{sj}(k)$ cannot associate with the $q$th target.

It is assumed that each target can produce only one correct measurement at a certain time, and each measurement can correspond to only one target. Therefore, in order to correctly represent the association between measurements and targets in each feasible event, it is necessary to split the validation matrix. The split result of validation matrix is called hypothesis matrix. Based on the above assumption, each column of the hypothesis matrix can have one 1, and each row of the hypothesis matrix can also have one 1. Let $A_{si}(k)$ represents the $i$th feasible event, then the hypothesis matrix can be written as:

$$\hat{\mathbf{\Omega}}_s(A_{si}(k)) = \begin{bmatrix} \hat{\omega}^1_{1i}(A_{si}(k)) & \cdots & \hat{\omega}^Q_{1i}(A_{si}(k)) \\ \vdots & \cdots & \vdots \\ \hat{\omega}^1_{m_{sk}i}(A_{si}(k)) & \cdots & \hat{\omega}^Q_{m_{sk}i}(A_{si}(k)) \end{bmatrix}, \tag{13}$$

where

$$\hat{\omega}^q_{ji}(A_{si}(k)) = \begin{cases} 1, & A^q_{sji}(k) \subset A_{si}(k) \\ 0, & otherwise. \end{cases} \tag{14}$$

In the above equation, $A^q_{sji}(k)$ represents the event that $j$th non-coherent measurement is associated with the $q$th target in the $i$th feasible event. That is, only when $A^q_{sji}(k)$ occurs in the $i$th feasible event $A_{si}(k)$, the corresponding element $\hat{\omega}^q_{ji}(A_{si}(k))$ in the hypothesis matrix is 1, otherwise $\hat{\omega}^q_{ji}(A_{si}(k))$ is 0. In all feasible events, the event that the $j$th non-coherent measurement is associated with the $q$th target is denoted as $A^q_{sj}(k)$. Clearly,

$$A^q_{sj}(k) = \bigcup_{i=1}^{n_{sk}} A^q_{sji}(k), \tag{15}$$

where $n_{sk}$ is the total number of feasible events in non-coherent scenario. Then the probability of association between the $j$th non-coherent measurement and the $q$th target is:

$$\begin{aligned} \beta^q_{sj}(k) &= \Pr\left(A^q_{sj}(k)\Big|\mathbf{Z}_s(k), \mathbf{Z}^{k-1}_u\right) = \Pr\left(\bigcup_{i=1}^{n_{sk}} A^q_{sji}(k)\Big|\mathbf{Z}_s(k), \mathbf{Z}^{k-1}_u\right) \\ &= \sum_{i=1}^{n_{sk}} \hat{\omega}^q_{ji}(A_{si}(k)) \Pr\left(A_{si}(k)\Big|\mathbf{Z}_s(k), \mathbf{Z}^{k-1}_u\right) \\ &= \sum_{i=1}^{n_{sk}} \hat{\omega}^q_{ji}(A_{si}(k)) \frac{\Pr\left(A_{si}(k), \mathbf{Z}_s(k), \mathbf{Z}^{k-1}_u\right)}{\Pr\left(\mathbf{Z}_s(k), \mathbf{Z}^{k-1}_u\right)} \\ &= \sum_{i=1}^{n_{sk}} \hat{\omega}^q_{ji}(A_{si}(k)) \frac{\Pr\left(A_{si}(k), \mathbf{Z}_s(k), \mathbf{Z}^{k-1}_u\right)\big/\Pr\left(\mathbf{Z}^{k-1}_u\right)}{\Pr\left(\mathbf{Z}_s(k), \mathbf{Z}^{k-1}_u\right)\big/\Pr\left(\mathbf{Z}^{k-1}_u\right)} \\ &= \sum_{i=1}^{n_{sk}} \hat{\omega}^q_{ji}(A_{si}(k)) \frac{\Pr\left(\mathbf{Z}_s(k)\Big|A_{si}(k), \mathbf{Z}^{k-1}_u\right) \Pr\left(A_{si}(k)\Big|\mathbf{Z}^{k-1}_u\right)}{\sum_{j=1}^{n_{sk}} \Pr\left(\mathbf{Z}_s(k)\Big|A_{sj}(k), \mathbf{Z}^{k-1}_u\right) \Pr\left(A_{sj}(k)\Big|\mathbf{Z}^{k-1}_u\right)}, \end{aligned} \tag{16}$$

where

$$\Pr\left(A_{si}(k)\Big|\mathbf{Z}^{k-1}_u\right) = \frac{1}{n_{sk}}, \tag{17}$$

$$\begin{aligned} \Pr\left(\mathbf{Z}_s(k)\Big|A_{si}(k), \mathbf{Z}^{k-1}_u\right) &= \prod_{j=1}^{m_{sk}} \Pr\left(z_{sj}(k)\Big|A_{si}(k), \mathbf{Z}^{k-1}_u\right) \\ &= \prod_{j=1}^{m_{sk}} \frac{1}{\sqrt{|2\pi\mathbf{S}^q_s(k)|}} \exp\left\{\begin{array}{l} -\frac{1}{2}\left[z_{sj}(k) - \hat{z}^q_u(k|k-1)\right]^T \left(\mathbf{S}^q_s(k)\right)^{-1} \\ \times \left[z_{sj}(k) - \hat{z}^q_u(k|k-1)\right] \end{array}\right\}. \end{aligned} \tag{18}$$

In this subsection, the known measurements are $\mathbf{Z}_s(k)$, $\mathbf{Z}^{k-1}_u$, and $\mathbf{Z}^{k-1}_s$. Since the accuracy of coherent measurements $\mathbf{Z}^{k-1}_u$ is better than that of non-coherent measurements, the predication of the coherent measurements are used as the center of validation gate, namely, the form of (18). Moreover, (18) is the realization of the nearest neighbor algo-

rithm from the perspective of probability. The filtered state vector and the corresponding covariance matrix are:

$$\hat{x}_s^q(k|k) = \hat{x}_u^q(k|k-1) + \sum_{j=1}^{m_{sk}} \beta_{sj}^q(k) \mathbf{K}_s^q(k) \left( z_{sj}(k) - \hat{z}_u^q(k|k-1) \right), \tag{19}$$

$$\begin{aligned}
\mathbf{P}_s^q(k|k) = \mathbf{P}_u^q(k|k-1) &- \mathbf{K}_s^q(k)\mathbf{S}_s^q(k)\left(\mathbf{K}_s^q(k)\right)^T + \\
\sum_{j=1}^{m_{sk}} \beta_{sj}^q \Big( \hat{x}_u^q(k|k-1) &+ \mathbf{K}_s^q(k)\left( z_{sj}(k) - \hat{z}_u^q(k|k-1) \right) \Big) \times \\
\Big( \hat{x}_u^q(k|k-1) &+ \mathbf{K}_s^q(k)\left( z_{sj}(k) - \hat{z}_u^q(k|k-1) \right) \Big)^T \\
- \hat{x}_s^q(k|k) &\left( \hat{x}_s^q(k|k) \right)^T,
\end{aligned} \tag{20}$$

respectively. According to (19), the probability of association $\beta_{sj}^q(k)$ is used to weight the filtering results to obtain the correct association between the non-coherent measurements and the targets. Meanwhile, the prediction of coherent measurement is used to improve the accuracy of non-coherent filtering results.

### 2.2.2. Coherent JPDAF

In Section 2.2.1, non-coherent filtering results are obtained based on the coherent measurements $\mathbf{Z}_u^{k-1}$ and non-coherent measurements $\mathbf{Z}_s^k$. Therefore, the ambiguity of new coherent measurements $\mathbf{Z}_u(k)$ can be resolved based on the non-coherent filtering results in this subsection. At time $k-1$, it is known that the filtering result of the $q$th target state is $\hat{x}_s^q(k|k)$, and the corresponding covariance matrix is $\mathbf{P}_s^q(k|k)$. Then, the prediction of measurement and the covariance matrix are:

$$\begin{aligned}
\hat{z}_s^q(k|k) &= \mathbf{H}(k)\hat{x}_s^q(k|k), \\
\mathbf{S}_u^q(k) &= \mathbf{H}(k)\mathbf{P}_s^q(k|k)\mathbf{H}^T(k) + \mathbf{R}_u^q(k),
\end{aligned} \tag{21}$$

respectively. The corresponding gain matrix is:

$$\mathbf{K}_u^q(k) = \mathbf{P}_s^q(k|k)\mathbf{H}^T(k)\left(\mathbf{S}_u^q(k)\right)^{-1}. \tag{22}$$

Because of the ambiguity in coherent measurements, the number of coherent measurements satisfies $m_{uk} > m_{sk}$. Similar to the definition in Section 2.2.1, in the coherent scenario, the validation matrix and the hypothesis matrix are:

$$\mathbf{\Omega}_u = \begin{bmatrix} \omega_1^1 & \cdots & \omega_1^Q \\ \vdots & \cdots & \vdots \\ \omega_{m_{uk}}^1 & \cdots & \omega_{m_{uk}}^Q \end{bmatrix}, \tag{23}$$

$$\hat{\mathbf{\Omega}}_u(A_{ui}(k)) = \begin{bmatrix} \hat{\omega}_{1i}^1(A_{ui}(k)) & \cdots & \hat{\omega}_{1i}^Q(A_{ui}(k)) \\ \vdots & \cdots & \vdots \\ \hat{\omega}_{m_{uk}i}^1(A_{ui}(k)) & \cdots & \hat{\omega}_{m_{uk}i}^Q(A_{ui}(k)) \end{bmatrix}, \tag{24}$$

respectively. In (24), $A_{ui}(k)$ represents the $i$th feasible event in the coherent scenario, and $n_{uk}$ is the total number of feasible events. Then, the probability of association between the $j$th coherent measurement and the $q$th target is:

$$
\begin{aligned}
\beta_{uj}^q(k) &= \Pr\left(A_{uj}^q(k)\middle|\mathbf{Z}_u(k),\mathbf{Z}_s(k),\mathbf{Z}^{k-1}\right) = \Pr\left(\bigcup_{i=1}^{n_{uk}} A_{uji}^q(k)\middle|\mathbf{Z}_u(k),\mathbf{Z}_s(k),\mathbf{Z}^{k-1}\right) \\
&= \sum_{i=1}^{n_{uk}} \hat{\omega}_{ji}^q(A_{ui}(k))\Pr\left(A_{ui}(k)\middle|\mathbf{Z}_u(k),\mathbf{Z}_s(k),\mathbf{Z}^{k-1}\right) \\
&= \sum_{i=1}^{n_{uk}} \hat{\omega}_{ji}^q(A_{ui}(k))\frac{\Pr\left(A_{ui}(k),\mathbf{Z}_u(k),\mathbf{Z}_s(k),\mathbf{Z}^{k-1}\right)}{\Pr\left(\mathbf{Z}_u(k),\mathbf{Z}_s(k),\mathbf{Z}^{k-1}\right)} \\
&= \sum_{i=1}^{n_{uk}} \hat{\omega}_{ji}^q(A_{ui}(k))\frac{\Pr\left(A_{ui}(k),\mathbf{Z}_u(k),\mathbf{Z}_s(k),\mathbf{Z}^{k-1}\right)\Big/\Pr\left(\mathbf{Z}_s(k),\mathbf{Z}^{k-1}\right)}{\Pr\left(\mathbf{Z}_u(k),\mathbf{Z}_s(k),\mathbf{Z}^{k-1}\right)\Big/\Pr\left(\mathbf{Z}_s(k),\mathbf{Z}^{k-1}\right)} \\
&= \sum_{i=1}^{n_{uk}} \hat{\omega}_{ji}^q(A_{ui}(k))\frac{\Pr\left(\mathbf{Z}_u(k)\middle|A_{ui}(k),\mathbf{Z}_s(k),\mathbf{Z}^{k-1}\right)\Pr\left(A_{ui}(k)\middle|\mathbf{Z}_s(k),\mathbf{Z}^{k-1}\right)}{\sum\limits_{j=1}^{n_{uk}}\Pr\left(\mathbf{Z}_u(k)\middle|A_{uj}(k),\mathbf{Z}_s(k),\mathbf{Z}^{k-1}\right)\Pr\left(A_{uj}(k)\middle|\mathbf{Z}_s(k),\mathbf{Z}^{k-1}\right)},
\end{aligned}
\tag{25}
$$

where

$$
\Pr\left(A_{ui}(k)\middle|\mathbf{Z}_s(k),\mathbf{Z}^{k-1}\right) = \frac{1}{n_{uk}},
\tag{26}
$$

$$
\begin{aligned}
\Pr\left(\mathbf{Z}_u(k)\middle|A_{ui}(k),\mathbf{Z}_s(k),\mathbf{Z}^{k-1}\right) &= \prod_{j=1}^{m_{uk}}\Pr\left(\mathbf{z}_{uj}(k)\middle|A_{ui}(k),\mathbf{Z}_s(k),\mathbf{Z}^{k-1}\right) \\
&= \prod_{j=1}^{m_{uk}}\frac{1}{\sqrt{|2\pi\mathbf{S}_u^q(k)|}}\exp\left\{-\frac{1}{2}\left[\mathbf{z}_{uj}(k)-\hat{\mathbf{z}}_s^q(k|k)\right]^T\left(\mathbf{S}_u^q(k)\right)^{-1}\left[\mathbf{z}_{uj}(k)-\hat{\mathbf{z}}_s^q(k|k)\right]\right\}.
\end{aligned}
\tag{27}
$$

In this subsection, the known measurements are $\mathbf{Z}_u(k)$, $\mathbf{Z}_s(k)$, and $\mathbf{Z}^{k-1}$. The filtering results of non-coherent measurements $\hat{\mathbf{z}}_s^q(k|k)$ are used in this subsection to resolve the ambiguity of coherent measurements $\mathbf{Z}_u(k)$, $\hat{\mathbf{z}}_s^q(k|k)$ is used as the center of validation gate, as shown in (27). Then, the filtered state vector and the corresponding covariance matrix are:

$$
\hat{\mathbf{x}}_u^q(k|k) = \hat{\mathbf{x}}_s^q(k|k) + \sum_{j=1}^{m_{uk}}\beta_{uj}^q(k)\mathbf{K}_u^q(k)\left(\mathbf{z}_{uj}(k)-\hat{\mathbf{z}}_s^q(k|k)\right),
\tag{28}
$$

$$
\begin{aligned}
\mathbf{P}_u^q(k|k) = {}& \mathbf{P}_s^q(k|k) - \mathbf{K}_u^q(k)\mathbf{S}_u^q(k)\left(K_u^q(k)\right)^T + \\
& \sum_{j=1}^{m_{uk}}\beta_{uj}^q\left(\hat{\mathbf{x}}_s^q(k|k)+\mathbf{K}_u^q(k)\left(\mathbf{z}_{uj}(k)-\hat{\mathbf{z}}_s^q(k|k)\right)\right)\times \\
& \left(\hat{\mathbf{x}}_s^q(k|k)+\mathbf{K}_u^q(k)\left(\mathbf{z}_{uj}(k)-\hat{\mathbf{z}}_s^q(k|k)\right)\right)^T \\
& -\hat{\mathbf{x}}_u^q(k|k)\left(\hat{\mathbf{x}}_u^q(k|k)\right)^T,
\end{aligned}
\tag{29}
$$

respectively. From (28), this subsection further realizes coherent JPDAF based on the non-coherent JPDAF results in the previous subsection. The results of non-coherent JPDAF are used to solve the problem of spatial crossing targets, and the ambiguity problem can be solved by using the non-coherent and coherent alternating JPDAF.

## 3. Results

In this section, the proposed NCCAF algorithm is simulated and verified. The simulation parameters are shown in Table 1.

**Table 1.** Simulation Parameters.

| Parameter Name | Parameter Value |
|---|---|
| Carrier frequency | 1 GHz |
| Bandwidth | 100 MHz |
| Number of targets | 2 |
| Initial target position | $(0, 0)$, $(7500\,\text{m}, 0)$ |
| Initial target velocity | $(150\,\text{m/s}, 150\,\text{m/s})$, $(-150\,\text{m/s}, 150\,\text{m/s})$ |
| Target acceleration | $(3\,\text{m/s}^2, -3\,\text{m/s}^2)$, $(-3\,\text{m/s}^2, -3\,\text{m/s}^2)$ |
| Time of tracking | 50 s |
| Interval of sampling time | 0.5 s |
| Standard deviation of non-coherent measurement | 10 m |
| Standard deviation of coherent measurement | 1 m |
| Number of Monte Carlo simulations | 100 |
| Signal-to-noise ratio | 20 dB |

The position RMSE is used to evaluate filtering performance, as shown below:

$$\text{RMSE}_{uPOS}(k) = \sqrt{\frac{1}{MC} \sum_{m=1}^{MC} \sum_{q=1}^{Q} \left[ \hat{x}_{um}^q(k|k) - x(k) \right]^2 + \left[ \hat{y}_{um}^q(k|k) - y(k) \right]^2}, \qquad (30)$$

where $MC$ represents the number of Monte Carlo simulations; $(\hat{x}_{um}^q(k|k), \hat{y}_{um}^q(k|k))$ represents the filtering value of the coherent measurement at time $k$; $(x(k), y(k))$ represents the target real position at time $k$. The difference between position RMSE in non-coherent scenario and the above formula is that $(\hat{x}_{um}^q(k|k), \hat{y}_{um}^q(k|k))$ is replaced with $(\hat{x}_{sm}^q(k|k), \hat{y}_{sm}^q(k|k))$.

Based on the above simulation parameters, the tracking results of multiple targets in spatial crossing are shown in Figure 2.

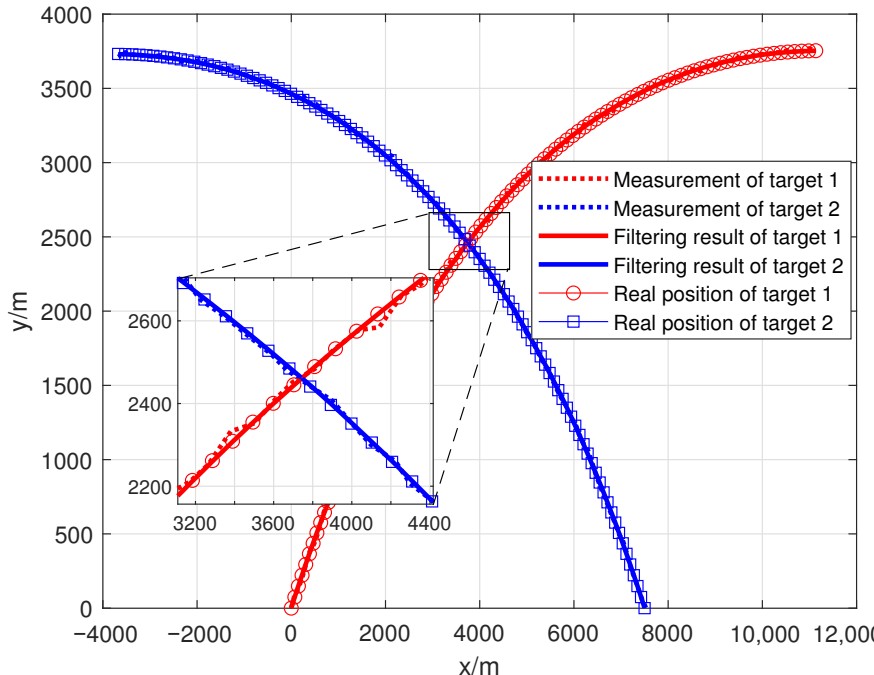

**Figure 2.** Topology of multi-target tracking.

Figure 2 shows the location of the target, and the unit of horizontal and vertical coordinates is meters. The red dotted line and blue dotted line in the figure represent the measured positions of target 1 and target 2 with time change, respectively. The red solid line and blue solid line represent the filtering results of the proposed tracking algorithm for target 1 and target 2, respectively. The red line with circle markers represents the real position of target 1. The blue line with square markers represents the real position of target 2. It can be seen from Figure 2 that the proposed NCCAF algorithm achieves stable tracking of two targets, and the filtered results are closer to the real positions of the targets than the measurement results.

The RMSE results of the tracking algorithm is shown in Figure 3. The horizontal coordinate in the figure is time, and the unit of it is second; the vertical coordinate is the position RMSE. The black solid line and black dotted line in the figure represent the RMSE results of target 1 and target 2 localization by data fusion, respectively. Data fusion here means that the parameter estimation result of target position based on target echo signal, which do not use the filtering algorithm. The blue solid line and blue dotted line represent the RMSE results of target 1 and target 2 localization by non-coherent JPDAF, respectively. The red solid line and the red dotted line represent the RMSE results of target 1 and target 2 localization by the proposed NCCAF algorithm, respectively.

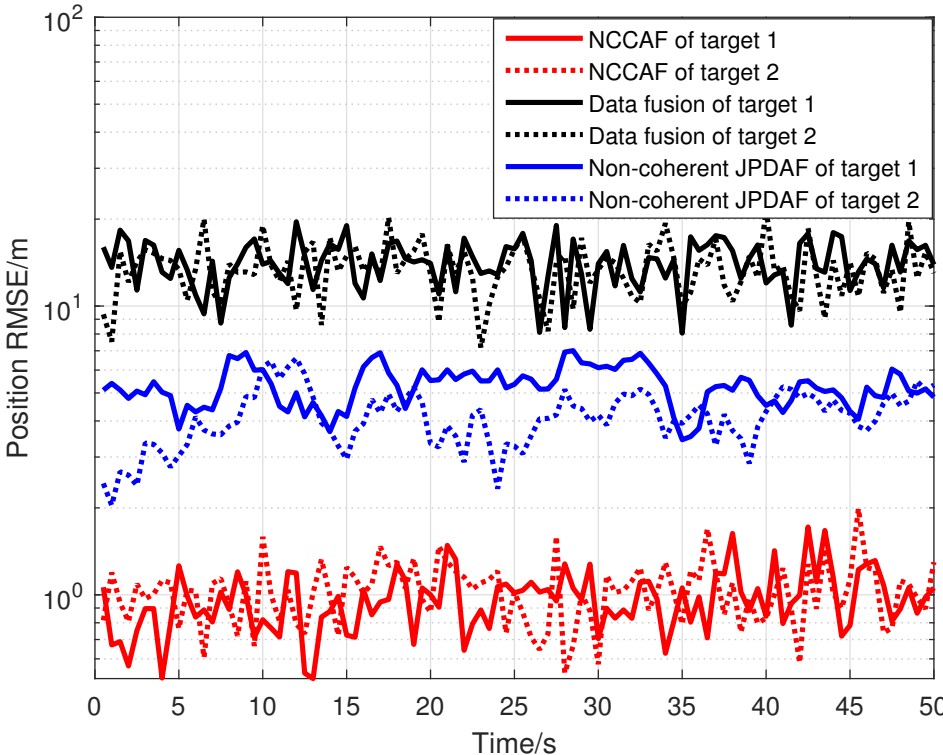

**Figure 3.** The root mean square error (RMSE) results of the tracking algorithm.

By comparing the black lines and the blue lines in Figure 3, it can be seen that the filtering algorithm has better RMSE results than the simple parameter estimation. This is because the simple parameter estimation only uses the echo signal of one coherent processing interval (CPI), while the filtering algorithm uses the echo signal of multiple CPI to improve the RMSE results. By comparing the blue lines and the red lines, it can be seen that the NCCAF algorithm proposed in this paper has better filtering accuracy than the non-coherent JPDAF in a multi-target scenario. This is because the NCCAF uses the result of non-coherent JPDAF to resolve the ambiguity of coherent measurements, thus ensuring the correct use of the coherent measurements to obtain good estimation performance.

## 4. Conclusions

Because of widely separated antennas of MIMO radar, much target position ambiguity will occur in high resolution mode, which brings difficulties to the realization of high resolution target localization. In order to make full use of all useful information about the MIMO radar, this paper proposes a two time filtering method. Based on the fusion measurement, the non-coherent JPDAF is carried out. Then, based on the non-coherent filtering result, the coherent JPDAF is proposed to achieve the high accuracy and stable tracking of multiple targets. The theoretical research and simulation verification in this paper provide a new solution for high accuracy ambiguity-free target localization and high accuracy stable multi-target tracking.

The algorithm proposed in this paper adopts two-step JPDAF, so the computational complexity is twice that of the JPDAF algorithm. After the JPDAF algorithm splits the validation matrix into several hypothesis matrices, the association weight is calculated based on the elements of each hypothesis matrix [31]. Therefore, compared to the Kalman filter, the main computational cost of the JPDAF algorithm comes from the number of hypothesis matrices. Taking coherent JPDAF as an example, when the validation matrix is an all-ones matrix, the number of hypothesis matrices reaches the maximum value $C^1_{m_{uk}} C^1_{m_{uk}-1} \cdots C^1_{m_{uk}-Q+1} = A^{m_{uk}-Q+1}_{m_{uk}}$. In view of the high computational cost of JPDAF caused by matrix splitting, Refs. [32–34] have proposed algorithms to optimize it. These algorithms can be applied to the algorithm proposed in this paper to reduce the computational cost.

In the research area of the high resolution mode of MIMO radar with widely separated antennas, Ref. [35] demonstrates the relation between radar locations, target location and localization accuracy. Hence, in the future, MIMO radar configuration optimization and the proposed algorithm in this paper will be combined to resolve the ambiguity of the high resolution mode of MIMO radar.

**Author Contributions:** Conceptualization, J.L.; data curation, J.L.; methodology, J.L. and F.L.; writing—original draft, Q.L. and H.L.; writing—review and editing, F.L. and Q.L. All authors have read and agreed to the published version of the manuscript.

**Funding:** This work was supported by the National Natural Science Foundation of China (Grant No. 62071045 and Grant No. 61625103).

**Institutional Review Board Statement:** Not applicable.

**Informed Consent Statement:** Not applicable.

**Data Availability Statement:** Not applicable.

**Acknowledgments:** The authors would like to thank the editor and anonymous reviewers for their helpful comments and suggestions.

**Conflicts of Interest:** The authors declare no conflict of interest.

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
