# Peer review of "Target Localization Based on High Resolution Mode of MIMO Radar with Widely Separated Antennas"

_remotesensing, doi:10.3390/rs14040902_

Round 1

Reviewer 1 Report

The referee is not very qualified to review the paper; however as a generic reader I notice that after an accurate introduction, starting from the section 2 the presentation is not well contextualized. All equations are introduced without adequate explanations or quotes to help the reader. (All references are in section 1; and that's it: not a single reference along the paper and even in the conclusion). 

Reviewer 2 Report

This paper focuses on utilizing JPDAF algorithm to gain high accuracy localization. The technique is well described and interesting results are demonstrated. However, there are some points improvement is needed as below.

  • The compared result of Fig.2 is not described enough, such as how filtered results are different.
  • Also, there is no mention about processing time whether the proposed method requires more computational cost.

Reviewer 3 Report

The authors of the paper developed algorithms for multi-input and multi-output radar (MIMO) signal processing based on the procedure described in literature 29. By using the state equation and measurement equation, the echo data of multiple coherent processing interval is collected to improve the target localization accuracy as much as possible. Non-coherent joint probability data association filer (JPDAF) is used to achieve stable tracking of spatial cross targets without ambiguity measurements. Using of numerical simulation by means of the non-coherent JPDAF and coherent alternating JPDAF (NCCAF) algorithms accuracy localization of multiple targets was verified. A very simple target movement model was used to verify accuracy. In the future, it would be useful to validate algorithms using a more complex target movement model. It is also not clear why a speed of 10 m.s-1 and small values variance of non-coherent and coherent measurement was chosen. The conclusion of the paper should be supplemented by a comparison of the obtained results with other works in this area of research. I consider the results of the authors' research to be a good input into the issue of MIMO radar signal processing. I recommend publishing the submitted article after a small revision.
